# Superior Gait Symmetry and Postural Stability among Yoga Instructors—Inertial Measurement Unit-Based Evaluation

**DOI:** 10.3390/s22249683

**Published:** 2022-12-10

**Authors:** Ang-Chieh Lin, Tzu-Tung Lin, Yin-Keat Tan, Wei-Ren Pan, Chih-Jen Shih, Chun-Ju Lee, Szu-Fu Chen, Fu-Cheng Wang

**Affiliations:** 1Department of Physical Medicine and Rehabilitation, Cheng Hsin General Hospital, Taipei 112, Taiwan; 2Department of Mechanical Engineering, National Taiwan University, Taipei 10617, Taiwan; 3Yogabija Studio, Taipei 106, Taiwan; 4Department of Physiology and Biophysics, National Defense Medical Center, Taipei 114, Taiwan

**Keywords:** gait, symmetry, balance, yoga, IMU, single-leg stance

## Abstract

This study investigates gait symmetry and single-leg stance balance of professional yoga instructors versus age-matched typically developed controls using inertial measurement unit (IMU)-based evaluation. We recruited twenty-five yoga instructors and twenty-five healthy control subjects to conduct the walking experiments and single-leg stance tests. Kinematic data were measured by attaching IMUs to the lower limbs and trunk. We assessed the asymmetry of swing phases during the normal-walk and tandem-walk tests with eyes open and closed, respectively. The subjects subsequently conducted four single-leg stance tests, including a single-leg stance on both legs with eyes open and closed. Two balance indexes regarding the angular velocities of the waist and chest were defined to assess postural stability. The gait asymmetry indexes of yoga instructors were significantly lower than those of the typically developed controls. Similarly, the yoga instructors had better body balance in all four single-leg stance tests. This study’s findings suggest that yoga improves gait asymmetry and balance ability in healthy adults. In the future, further intervention studies could be conducted to confirm the effect of yoga training.

## 1. Introduction

Maintaining a stable gait depends on core and lower limb muscle strength, postural control, and peripheral sensation [1,2]. Examples include the ability to maintain position, respond to voluntary body and extremity movements, and react to external disturbances [3]. Reduced gait stability is associated with an increased risk of falling [4], diminished mobility [5], and cognition decline [6]. Substantial evidence demonstrates that exercise interventions, such as muscle strength, endurance, and balance training, contribute to an increase in gait stability in healthy adults, especially older adults [7].

Yoga is a physical activity that includes the practice of specific physical postures, breathing techniques, and meditation [8]. Growing evidence shows that yoga can improve balance, strength, and flexibility in the general population and in older adults with high levels of enjoyment, which increases exercise adherence [8]. Yoga practice consists of complex movements involving 3-dimensional motions, including proper body posture, alignment, and rhythm, to maintain static and dynamic balance. A systematic review and meta-analysis showed that yoga practice resulted in minor improvements in balance and medium improvements in physical mobility among the population aged >60 years [9]. Apart from the general population and older adults, previous studies have shown that yoga can improve mobility, strength, balance, and gait stability in patients with neurological and musculoskeletal disorders [10,11,12,13,14].

Previous studies which evaluated yoga benefits over gait performance and balance mainly based on clinical scales, such as the Expanded Disability Status Score [11], Berg Balance Scale [11,12], and Short Physical Performance Battery [9]. However, these are subjective parameters that lack an objective comparison basis. On the other hand, some devices may provide more objective information, e.g., dynamic posturography, optical motion capture systems, and force plates. However, these devices are often unaffordable when testing gait and balance performance in a community or clinical setting due to the limitations of laboratory-based settings. Therefore, this paper applies inexpensive wearable devices to provide an objective evaluation of yoga benefits.

Human gait is symmetrical and rhythmically periodic. A gait cycle typically consists of approximately 60% stance and 40% swing phases, and the swing time of the two legs is usually symmetrical for healthy individuals. Therefore, the symmetry of the swing time of two legs can be considered to estimate gait stability. Multiple methods have been proposed for measuring gait data. For instance, Wong et al. [15] applied load sensors to analyze foot contact patterns and evaluate walking ability. Bilro et al. [16] used wearable sensors to monitor gait. In the current study, inertial measurement unit (IMU), a wearable device able to record 3-axis accelerometers, 3-axis gyroscopes, and 3-axis magnetometers, was used to measure kinematic data and to evaluate gait behaviors and postural balance during normal walking, tandem walking, and single-leg stance. IMU sensors have been previously validated as a viable alternative to camera-based motion capture, which can be used to measure multiple aspects of balance and gait in isolation to gather data from individuals with movement disorders [17]. To date, studies have yet to report the application of wearable IMUs to quantitatively evaluate the effects of yoga on gait performance and balance.

The single-leg stance is a frequently used clinical tool for assessing balance in individuals with various balance disorders [18]. The single-leg stance test is an easy clinical balance test that can assess postural steadiness in a relatively static condition. Poor single-leg balance is a significant predictor of injurious falls, especially in older adults [19]. Because vision also plays a crucial role in processing and integrating other sensory information involving postural control and balance [20], the impact of visual feedback can be avoided by conducting a single-leg stance test with eyes both closed and open.

This study aimed to investigate professional yoga instructors’ gait performance and balance ability versus age-matched typically developed controls. We applied the IMUs to measure their kinematic data during normal walking, tandem walking, and single-leg stance.

## 2. Materials and Methods

### 2.1. Participants

All gait and balance performance data were collected from participants who provided written informed consent, and permission was obtained for all photographs used in this manuscript. We recruited twenty-five experienced yoga instructors and twenty-five age/sex-matched control individuals without yoga activity to conduct the walking and single-leg stance experiments, as shown in Figure 1. The inclusion criteria of the yoga instructors were as follows: (a) overall good health and reasonable cooperation with instructions, (b) between 25 to 60 years old, (c) having the minimum RYT-200 (Registered Yoga Teacher-200 h of yoga education) certification with the Yoga Alliance, (d) practicing three to six days per week and at least ten years of experience in yoga, (e) no limb or leg discrepancy, (f) no history of surgery on the lower limbs or spine, and (g) no history of musculoskeletal injury over the lower back in the past six months. The non-yoga control subjects were age- and sex-matched healthy participants who had no experience with yoga training and were free from injury (no musculoskeletal injuries or neurological conditions). The non-yoga control subjects participated in regular exercise two or three times a week.

### 2.2. Demographic Data

The study group comprised 25 yoga instructors (20 women and 5 men) and 25 typically developed controls (21 women and 4 men). The basic data of all subjects are illustrated in Appendix A. The mean ages (range) of the patients in these two groups were 44.6 ± 7.9 (28–59) years and 44.5 ± 8.7 (26–59) years, respectively. Details of the participants’ demographic data are presented in Table 1. No significant differences were observed between the yoga instructors and the healthy control groups.

### 2.3. The IMU System

We applied the OPAL IMU system with wearable IMUs [21] to measure the kinematic data of the subjects during the experiments. The specifications of the OPAL IMU system are listed in Table 2. Four IMUs were attached to the subjects: two on the middle of the shanks at midpoint between patellar level and lateral malleolus level, one on the waist at anterior superior iliac spine (ASIS) level, and one on the chest at scapular spine level, as shown in Figure 2. These IMUs recorded the kinematic data of the subjects at a sampling rate of 128 Hz [22].

### 2.4. Evaluation of Gait Performance

Gait performance was assessed by walking tests; all subjects walked in a straight line at their most comfortable pace, as shown in Figure 1a. They were then required to perform tandem walking on a straight line for 20 m, where the front foot’s heel touched the toe of the standing foot, with eyes open and closed, respectively. They were required to perform this walk 2 times. Their arms were crossed in front of their chest (see Figure 1b,c) to minimize upper limb swaying, control potential compensatory strategies to maintain balance and standardize our testing. Two IMUs were attached to their shanks to measure kinematic data during the walking tests.

A gait cycle typically consists of seven events: heel strike, loading respond, mid-stance, terminal stance, pre-swing, mid-swing and terminal swing. In this study, we focus on three important gait events, as shown in Figure 3a: mid-swing (MS), heel strike (HS), and toe-off (TO). The maximal angular velocity of the shank occurs at MS at the maximum angular velocity of the shank in the sagittal plane during the gait cycle. HS is when the heel touches the ground, where the angular velocity of the shank has the first negative trough after the MS. TO is when the toes leave the ground, where the angular velocity of the shank has the last negative trough before the MS. The stance phase is defined as the time interval from HS to TO and accounts for about 60% of the gait cycle. The swing phase takes about 40% of the gait cycle and is defined as the time interval from TO to the next HS.

We measured the subjects’ angular velocities of the shanks. For example, the angular velocity changes within gait cycles of a yoga instructor and a healthy control are shown in Figure 3b,c. We superimpose the angular velocity changes in each gait cycle, as illustrated in Figure 3d,e.

The asymmetry of the swing phases in each gait cycle was defined as follows [22]:$$Asym_{SP}=\frac{\left|{\left(P_{SW}\right)}_{R}-{\left(P_{SW}\right)}_{L}\right|}{\min\left\{\begin{array}{cc} {\left(P_{SW}\right)}_{R} & ,\;{\left(P_{SW}\right)}_{L} \end{array}\right\}}\times 100\%,$$
where ${\left(P_{SW}\right)}_{R}$ and ${\left(P_{SW}\right)}_{L}$ represent the percentage of swing phases on the right and left feet, respectively.

The ideal $Asym_{SP}$ should be zero because healthy individuals tend to have symmetrical gaits, where the swing phases are approximately 40% on both sides. Clinically, gait asymmetry has been reported to lead to several negative consequences, such as increased energy cost of locomotion, impaired balance control, and a higher dynamic load, with an increased risk of musculoskeletal injury to the dominant lower limb [24,25]. Thus, gait asymmetry is a relevant indicator for differentiating between normal and pathological gaits [26]. Hence, we can define the following index to quantify the subjects’ gait performance balance [22]:$$J_{Gait}={\left(\frac{1}{N}\sum_{i=1}^{N}{\left|Asym_{SP}(i)\right|}^{2}\right)}^{1/2}$$
where *N* is the number of gait cycles, and *i* represents the *i*-th gait cycle. $J_{Gait}$ represents the very extended root mean square index applied to $Asym_{SP}$.

### 2.5. Evaluation of Postural Stability

Postural stability was evaluated using single-leg stance tests, as shown in Figure 1d,e. The subjects underwent the following single-leg stance tests to investigate their balance ability:(1)Standing on the dominant foot with eyes open.(2)Standing on the dominant foot with eyes closed.(3)Standing on the non-dominant foot with eyes open.(4)Standing on the non-dominant foot with eyes closed.

The subjects stood on one foot at each action for approximately 60 s to quantify their balance ability, but they were free to stand on both feet if their body lost balance. Single-leg stance time of each individual is provided at Appendix B.

IMUs were implemented on the subjects’ waists and chests to record their angular velocities during actions. When we stand on one foot, our bodies must adjust to maintain a dynamic balance like an inverted pendulum. Therefore, we can define the following indexes to quantify the subjects’ balance abilities [22]:$$J_{bal}^{waist}=\frac{1}{N}\sum_{i=1}^{N}\left|\omega^{waist}(i)\right|=\frac{1}{N}\sum_{i=1}^{N}{\left({\left(\omega_{x}^{waist}(i)\right)}^{2}+{\left(\omega_{y}^{waist}(i)\right)}^{2}+{\left(\omega_{z}^{waist}(i)\right)}^{2}\right)}^{1/2}$$
$$J_{bal}^{chest}=\frac{1}{N}\sum_{i=1}^{N}\left|\omega^{chest}(i)\right|=\frac{1}{N}\sum_{i=1}^{N}{\left({\left(\omega_{x}^{chest}(i)\right)}^{2}+{\left(\omega_{y}^{chest}(i)\right)}^{2}+{\left(\omega_{z}^{chest}(i)\right)}^{2}\right)}^{1/2}$$
where *N* is the total number of samples. $\omega_{x}^{waist}(i),\omega_{y}^{waist}(i),\mathrm{and}\omega_{z}^{waist}(i)$ are the three-axial angular velocities from the waist IMU at the *i*-th sample. $\omega_{x}^{chest}(i),\omega_{y}^{chest}(i),\mathrm{and}\omega_{z}^{chest}(i)$ are the three-axial angular velocities from the chest IMU at the *i*-th sample. That is, $\omega^{chest}(i)$ and $\omega^{waist}(i)$ are the absolute angular velocities on the chest and the waist, respectively. The subject’s balance ability is better when the angular velocities are lower, indicating that the subject did not need much muscle force to maintain balance.

### 2.6. Statistical Analyses

The measured IMU data are illustrated in Appendix C. We applied the Shapiro-Wilk test to assess the distribution of data, and we presented the data as mean and standard deviation (mean ± SD). The overall difference between the groups was tested using Student’s t-test in the case of normal data distribution or nonparametric Mann–Whitney U test when the data were not normally distributed. Software for Microsoft Office Excel 2019 was used for IMU data analysis. Statistical significance was set at *p* < 0.05. A priori sample estimate was calculated using a large effect size (0.82) with the G*Power software (version 3.1; Kiel University, Kiel, Germany), and it was estimated that a minimum of 25 participants per group would be required to detect group difference in gait asymmetry using an alpha of 0.05 and a beta of 0.2.

## 3. Results

### 3.1. Evaluation of Gait Performance

We first evaluated gait performance by assessing the asymmetries of the swing phase $Asym_{SP}$ of yoga instructors and healthy controls. Gait symmetry has been assumed in healthy individuals who show minimal laterality with only subtle differences between the dominant and non-dominant legs. The gait data of all subjects are illustrated in Appendix D. Appendix E illustrates the gait performance index $J_{Gait}$ of all subjects.

The statistical results are illustrated in Table 3 and Figure 4. The gait performance indexes $J_{Gait}$ of Yoga instructors were significantly lower than that of the healthy control in both the normal walk (5.87 ± 2.78 vs. 8.06 ± 4.17, *p* = 0.029) and open-eye tandem walk tests (19.74 ± 4.43 vs. 26.47 ± 11.93, *p* = 0.023). Analysis of the indexes $J_{Gait}$ also showed a smaller value in the yoga instructors (38.08 ± 13.50 vs. 47.24 ± 23.99, *p* = 0.090) when compared to the healthy control in the close-eye tandem gait test; however, it did not reach statistical significance. The gait asymmetrical indexes $J_{Gait}$ of yoga instructors were 72.83%, 74.57%, and 80.61% of those of the healthy controls in the normal walk, open-eye tandem, and close-eye tandem walk tests, respectively.

### 3.2. Evaluation of Postural Stability

We also implemented IMUs on the subjects’ waist and chest to estimate their balance ability by their kinematic IMU data during the one-leg stance tests. The IMU data of all subjects are illustrated in Appendix F. Appendix G illustrates the balance index $J_{bal}^{waist}$ and $J_{bal}^{chest}$ of all subjects.

The analyses of $J_{bal}^{waist}$ are shown in Table 4 and Figure 5. Figure 5a–d shows the subjects’ angular velocities on the waist, where the yoga instructors had better body balance (smaller angular velocities) in all four one-leg stance tests. The balance indexes $J_{bal}^{waist}$ of yoga instructors were 73.43%, 73.68%, 58.10%, and 60.26% of the healthy controls, with p-values of 0.051, 0.025, 0.004, and 0.008, respectively, between the two groups in the four stance tests. This indicates superior postural stability over the waist level of yoga instructors during the single-leg stance.

Similarly, the yoga instructors had significantly smaller angular velocities on the chest in all four one-leg stance tests, as illustrated in Table 4. Figure 5e–h shows the subjects’ angular velocities on the chest. The balance index $J_{bal}^{chest}$ of yoga instructors was 68.42%, 65.56%, 46.94%, and 38.19% of healthy controls in the four one-leg stance tests, respectively. The p-values between the two groups were 0.029, 0.038, 0.008, and 0.017 in the four single-stance tests, respectively, indicating superior postural stability over the chest level of yoga instructors during single-leg stance.

## 4. Discussion

This study compared gait symmetry and postural stability between yoga instructors and typically developed controls. IMUs were attached to the participants’ calf to determine the gait symmetry $J_{Gait}$, with results showing less gait asymmetry in yoga instructors during normal walking and open-eye tandem gait tests. Although not statistically significant, the gait asymmetry index was also lower in yoga instructors during the closed-eye tandem gait test. For the single-leg stance test, we applied IMUs to the participants’ chest and waist and evaluated the index of postural stability by $J_{bal}^{chest}$ and $J_{bal}^{waist}$, which showed significantly smaller angular velocities on both the chest and waist in all four one-leg stance tests (on both the right and left legs, with eyes either opened or closed). These results indicated that yoga instructors had better gait symmetry and body balance than the healthy control group, suggesting that gait performance and postural stability might be improved by yoga training.

Gait performance was assessed using the symmetry of the swing phases of the bilateral lower limbs during a gait cycle. Clinically, increased gait asymmetry is associated with important functional consequences, such as reduced walking speed, increased energy expenditure, increased joint and bodily degradation, and increased susceptibility to injuries and falls [24,25,27]. Our results demonstrated that the asymmetries of the swing phases of yoga instructors were smaller than those of healthy controls during normal walking and open-eye tandem gait tests. Previous studies investigated the effects of yoga training on gait performance. Eight-week Hatha yoga practice reduced self-reported falls and improved balance and gait performance as measured by the Berg Balance Scale, functional gait assessment, and dynamic gait index in older adults [28]. Twelve-week yoga practice improved gait speed, double support time, and instrumented timed-up-and-go test results in healthy pregnant females [29]. Moreover, a gentle Iyengar yoga program increased peak hip extension and stride length among older adults with reduced hip extension, which is a known risk factor for recurrent falls [30]. Yoga training focuses on structural alignment of the physical body by combining a series of sitting and standing postures and movements. During practice, core muscles, including the transverse abdominis, multifidus, rectus abdominis, erector spinae, and lower limb muscles, work in coordination to maintain proper body posture and alignment. Furthermore, many yoga poses are unilateral (e.g., tree posture) and may require maximal effort on the weight-bearing side, thus leading to muscular symmetry between the dominant and non-dominant legs. In addition to working muscle groups in the lower limbs, unilateral movements place more pressure on the core muscles to help balance the body. Another possible reason for yoga instructors’ gait superiority is the body awareness required for yoga practice. Yoga practice integrates physical postures, breath control, and meditation to refine a person’s body awareness and the presence of limbs in space. These techniques may help correct asymmetries. The significant improvements in proprioception following yoga training in patients with Parkinson’s disease support this proposition [31].

It is of particular interest that yoga instructors also showed better gait symmetry than that of the control group during the open-eye tandem gait test. Although the asymmetry index $J_{Gait}$ showed a smaller value in the yoga instructors than in the healthy controls in the close-eye tandem gait test, it did not reach statistical significance. The lack of significant differences was likely due to the small sample size of this study. During tandem walking, the body’s center of mass is projected onto a relatively small surface area; thus, participants encounter more difficulty in maintaining dynamic balance. Therefore, the superiority in balance may be more significant during this test than during normal walking. Moreover, with both arms crossed in front of the chest, the body’s torque during tandem walking cannot be compensated by upper limb movements. Core muscle function is critical for controlling gait balance during tandem walking. The hip abductor and adductor muscles also contribute significantly to postural control in the mediolateral direction [32]. Taken together, our findings suggest that yoga practice may improve gait symmetry by enhancing dynamic postural control and body awareness and strengthening the core and lower limb muscles.

The one-leg stance is a widely adopted clinical tool for assessing balance in individuals with various balance disorders [18]. We showed that yoga instructors had less trunk sway while conducting the one-leg stance test, indicating that yoga instructors have better postural stability than the controls. The effects of yoga on balance have been investigated in several studies. A recent randomized controlled trial among patients with postmenopausal osteoporosis demonstrated that adding tree pose (Vrksasana) to conventional exercise per day for 12 weeks improved both static and dynamic balance, as well as the tandem walk test [14]. Yoga intervention also produced significant improvements in balance in healthy older fallers [9], children [33], and patients with Parkinson’s disease [34]. However, only the Berg Balance Scale was used as an evaluation tool in most of these studies. Previously, the one-leg stance test was used to quantitatively assess balance by measuring the center of pressure displacement using force platforms, which are generally confined to motion analysis laboratories [35]. In this study, an instrumented one-leg stance test based on wearable inertial sensors was applied to measure angular velocities on the trunk. This inertial sensor-based assessment allows clinicians to easily perform instrumental evaluation of balance disorders in the clinical environment.

We applied two IMUs to the chest and waist. $J_{bal}^{chest}$ can reflect upper trunk stability, whereas $J_{bal}^{waist}$ measures lower trunk stability. The chief muscles of the core that function in the sagittal plane include the rectus abdominis, transverse abdominis, erector spinae, multifidus, gluteus maximus, and the hamstrings. Co-contraction of the muscles on the anterior and posterior aspects of the trunk increases intra-abdominal pressure and generates greater trunk force, thereby stabilizing the trunk. The gluteus medius and minimus assist in maintaining a level pelvis, and co-contraction with their contralateral counterparts stabilizes the lumbar spine [36]. We showed that yoga instructors had better stability at both chest and waist levels, suggesting superior core and lower limb muscle function following yoga training. We conducted the one-leg stance test with eyes opened and closed on both legs, as visual feedback or leg dominance may affect the reliability of the balance test [35]. The yoga instructors had significantly smaller angular velocities on both the chest and waist in all four one-leg stance tests. Vision is involved in the processes of maintaining balance, as well as in the vestibular and somatosensory systems. To minimize the impact of visual feedback, we conducted a one-leg stance test with both eyes closed and open. The yoga instructors had significantly better one-leg stance performance with eyes open, in a less challenging condition, and with eyes closed, in a more challenging condition. These results indicate that yoga instructors can maintain better postural stability without visual feedback. Indeed, previous work has shown that highly trained yoga practitioners rely more on internal vestibular and proprioceptive signals than on external visual cues in a multisensory integration perceptual task [37]. An 8-week yoga program significantly improved postural stability by strengthening somatosensory and vestibular responses in the visually impaired population [38]. Thus, yoga practice may improve the responses of vestibular and somatosensory systems involved in balance.

Our study has several limitations. First, the participants were predominantly female, which may limit the generalizability of the results to male yoga instructors. However, this reflects the fact that over 70% of yoga practitioners are female. Second, only static postural control was investigated to assess balance ability in the current study. Other aspects of postural control include dynamic and reactive postural control parameters. A multidimensional evaluation will provide a broader view of balance control among yoga instructors. Another limitation is that our study design was a cross-sectional observational study; therefore, further interventional research is necessary to clarify the efficacy of yoga practice. Larger randomized controlled trials are needed to confirm the findings of this study and to ascertain whether yoga practice can improve gait function and balance ability in the healthy population and in other patients with neurological or orthopedic disorders.

## 5. Conclusions

This study investigated the potential benefits of yoga in improving gait symmetry and postural stability. Gait symmetry and body movements during one-leg stance were assessed by IMUs, a quantitative and novel approach compared to previous studies, which are mainly based on the Berg Balance Score and questionnaires. The results showed that the gait asymmetry indexes of yoga instructors were significantly lower than those of the healthy controls on normal walk and open-eye tandem gait tests. The yoga instructors also had better postural stability at both the waist and chest levels in all four single-leg stance tests. These results suggest that yoga training may improve gait symmetry and postural stability. The possible contributing factors are enhanced dynamic postural control, strengthening of the core and lower limb muscles, and better somatosensory and vestibular responses. Further studies are warranted to confirm the effects of yoga training on gait and balance performance in healthy subjects and patients with gait disorders.

## Figures and Tables

**Figure 1 sensors-22-09683-f001:**
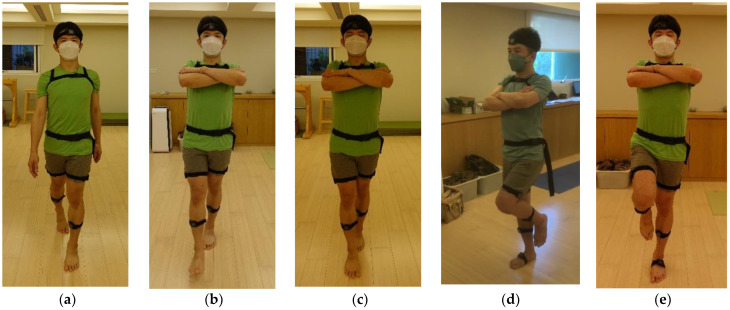
Experiments. (**a**) Normal walking tests; (**b**) Tandem gait test with eye open; (**c**) Tandem gait test with eye closed; (**d**) one-leg standing test with eye open; (**e**) one-leg standing test with eye closed.

**Figure 2 sensors-22-09683-f002:**
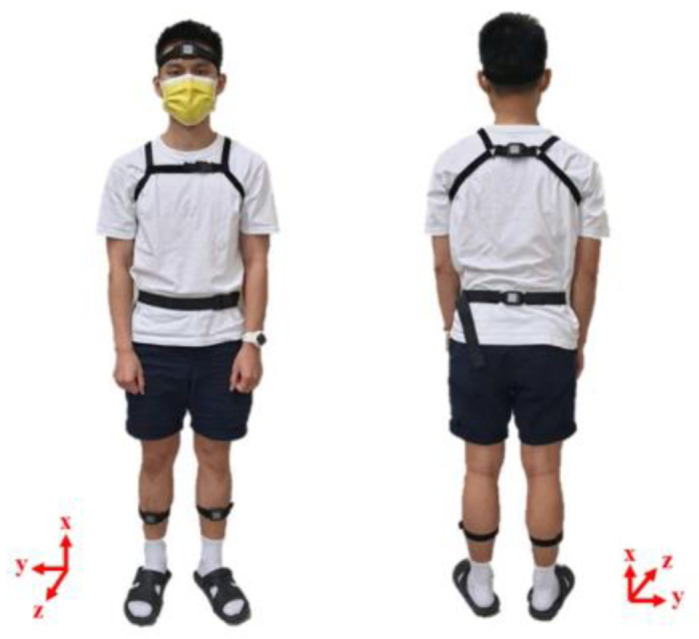
The IMU measurements.

**Figure 3 sensors-22-09683-f003:**
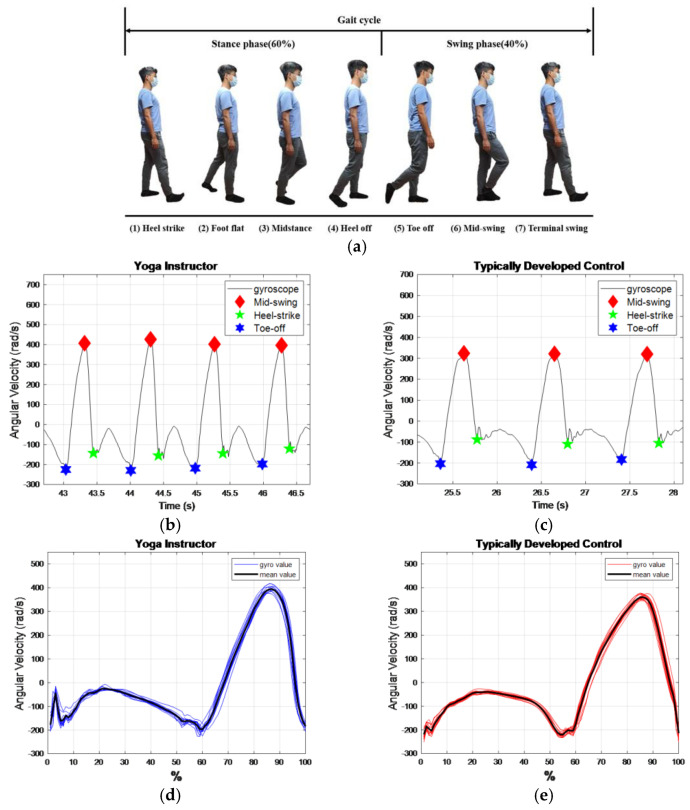
The gait data measurements. (**a**) A complete gait cycle; (**b**) the left gait response of a yoga instructor; (**c**) the right gait response of a healthy control; (**d**) the left gait cycle of a yoga instructor, ω_γ_ represents angular velocity recorded by IMU within a gait cycle; (**e**) the right gait cycle of a typically developed control.

**Figure 4 sensors-22-09683-f004:**
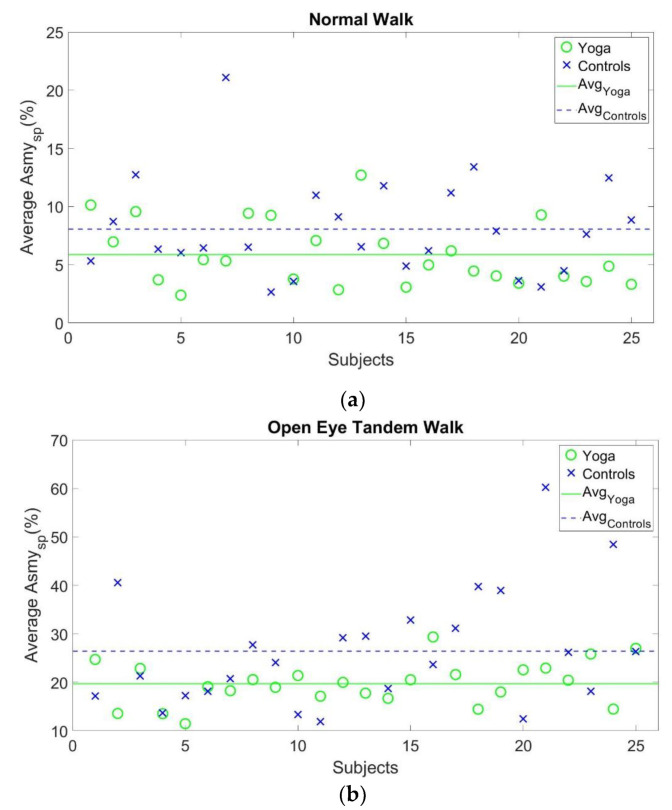
$J_{Gait}$ of subjects during the (**a**) normal walk, (**b**) open-eye tandem walk, (**c**) close-eye tandem walk tests.

**Figure 5 sensors-22-09683-f005:**
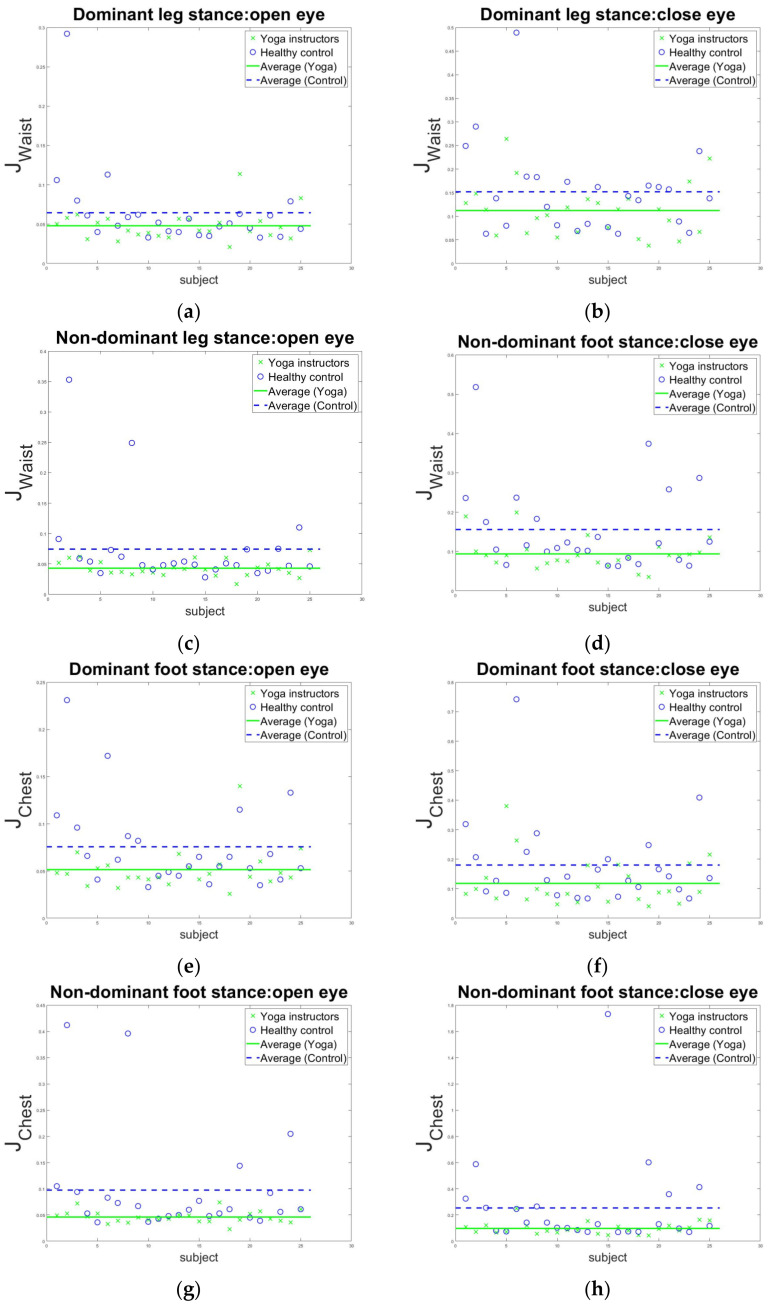
Indexes of the single-leg stance tests. $J_{bal}^{waist}$: (**a**) standing on the dominant foot with eyes open; (**b**) standing on the dominant foot with eyes closed; (**c**) standing on the non-dominant foot with eyes open; (**d**) standing on the non-dominant foot with eyes closed; $J_{bal}^{chest}$: (**e**) standing on the dominant foot with eyes open; (**f**) standing on the dominant foot with eyes closed; (**g**) standing on the non-dominant foot with eyes open; (**h**) standing on the non-dominant foot with eyes closed.

**Table 1 sensors-22-09683-t001:** Demographics of the subjects.

Characteristics	Yoga Instructor	Control Group
Age, years	44.6 ± 7.9	44.5 ± 8.7
Sex, no. (%)		
Male	5 (20%)	4 (16%)
Female	20 (80%)	21 (84%)
Weight, kg	54.9 ± 7.7	61.2 ± 14.5
Height, cm	163.2 ± 6.8	160.6 ± 7.8
BMI	20.6 ± 1.9	23.5 ± 3.6

**Table 2 sensors-22-09683-t002:** Specifications of the OPAL IMU system [21].

Variable	Value
Dimensions, mm	43.7 × 39.7 × 13.7
Weight (with battery), g	<25
Resolutions, bits	17.5
Sampling rates, Hz	20 to 128
Transmission range (line of sight), m	30
Ranges of the accelerometer, g	±200
Ranges of the gyroscope, deg/s	±2000
Ranges of the magnetometer, Gauss	±8
Lin’s concordance correlation coefficients	>0.93 [23]

**Table 3 sensors-22-09683-t003:** Analyses of gait performance in three walking tests.

$J_{Gait}$	Yoga Instructor		Control Group		
	Mean	SD	Mean	SD	*p*-Value
Normal walk	5.87	2.78	8.06	4.17	0.028
Tandem gait (eyes open)	19.74	4.43	26.47	11.93	0.022
Tandem gait (eyes closed)	38.08	13.50	47.24	23.99	0.090

**Table 4 sensors-22-09683-t004:** Analyses of balance abilities in four single-leg stance tests.

Balance Parameters	Yoga Instructor		Control Group		
$J_{bal}^{waist}$	Mean	SD	Mean	SD	*p*-Value
Dominant foot (eyes opened)	0.048	0.019	0.064	0.052	0.051
Dominant foot (eyes closed)	0.102	0.056	0.152	0.093	0.025
Non-dominant foot (eyes opened)	0.041	0.013	0.074	0.072	0.004
Non-dominant foot (eyes closed)	0.083	0.039	0.156	0.110	0.008
$J_{bal}^{chest}$	Mean	SD	Mean	SD	*p*-Value
Dominant foot (eyes opened)	0.052	0.022	0.076	0.047	0.029
Dominant foot (eyes closed)	0.120	0.080	0.018	0.146	0.038
Non-dominant foot (eyes opened)	0.046	0.012	0.098	0.099	0.008
Non-dominant foot (eyes closed)	0.097	0.047	0.254	0.345	0.017

## Data Availability

The dataset in this paper are available as follows: The IMU data of all subjects: http://140.112.14.7/~sic/PaperMaterial/IMU_data.zip, accessed on 10 November 2022. The gait cycles of all subjects: http://140.112.14.7/~sic/PaperMaterial/Gait Performance.zip, accessed on 10 November 2022. The IMU responses in the stance tests: http://140.112.14.7/~sic/PaperMaterial/Balance IMU data.zip, accessed on 10 November 2022.

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
