# Peer review of "Superior Gait Symmetry and Postural Stability among Yoga Instructors—Inertial Measurement Unit-Based Evaluation"

_sensors, 2022, doi:10.3390/s22249683_

Round 1

Reviewer 1 Report

The manuscript presents a study that demonstrates the better gait symmetry and single-leg stance balance in yoga instructors against healthy normal controls with similar anthropometric features. In opposition to other studies, the authors use wearable IMU sensors to measure the angular acceleration according to several anatomical planes in waist, chest, and shanks (through gyroscopes from a IMU commercial system for research, OPAL). This manner, the findings support the application of wearable IMU sensor to longitudinal studies and on-line frameworks.

The manuscript is well structured and written, the methodology is properly designed and presented, and results and discussion support the claimed findings. The following list of items summarizes several minor comments: 

  1. Introduction.

-          (Line 48) Which is the meaning of “Multiple dimensional motions”? Perhaps it could be substituted by three dimensional spatial motions or three dimensional and temporal spatial motions.

  1. Materials and methods

-          Format of reference [26] should be revised.

-          The relationship between the swing phase (Psw) variable used in equation (1) and figures 3 (d and e) should be clarified.

-          Correct “linically” term in line 165

-          The Jgait index is the very extended Root Mean Square index applied to Asym. Perhaps it could be pointed.

-          The minimum number of participants per group to detect group differences (statistical significance) should be justified (lines 201 – 202).

  1. Results

-          The anthropometric values of the study groups should be moved to the material and methods.

-          Table 3. The Jgait variable name should be added to the legend to clarify the values.

-          The titles of the four last plots in figure 5 (e – h) are incorrect. Their difference with respect to the previous plots is only the placement of the sensor. These titles must be corrected.

  1. Appendix B (gait performance data) and appendix D (balance IMU data) does not show data (angular accelerations’ samples) but the plots obtained from measured data. This must be clarified in the text. However, a better option could be to include the data samples in zip files, since this facilitates the reproducibility.

Author Response

Point 1: -          (Line 48) Which is the meaning of “Multiple dimensional motions”? Perhaps it could be substituted by three dimensional spatial motions or three dimensional and temporal spatial motions.

Response 1: Revised in the manuscript.

Point 2: 

  1. Materials and methods

-          Format of reference [26] should be revised.

-          The relationship between the swing phase (Psw) variable used in equation (1) and figures 3 (d and e) should be clarified.

-          Correct “linically” term in line 165

-          The Jgait index is the very extended Root Mean Square index applied to Asym. Perhaps it could be pointed.

The minimum number of participants per group to detect group differences (statistical significance) should be justified (lines 201 – 202).

Response 2: Revised in the manuscript.

Point 3: 

-          The anthropometric values of the study groups should be moved to the material and methods. (Response: Revised in the manuscript)

-          Table 3. The Jgait variable name should be added to the legend to clarify the values.(Response: Revised in the manuscript)

Point 3-3: Results - The titles of the four last plots in figure 5 (e – h) are incorrect. Their difference with respect to the previous plots is only the placement of the sensor. These titles must be corrected. 

Response 3-3: Thanks for reminding this fact. We want to confirm that figure 5  plot (a)-(d) represents data collected via the sensor attached to our participants' waist, while figure 5 plot (e)-(h) represents data   collected via the sensor attached to our participants' chest level at scapular spine. Therefore, the titles for our plots is based on the posture and recorded limb of our participant. 

Point 4: 

  1. Appendix B (gait performance data) and appendix D (balance IMU data) does not show data (angular accelerations’ samples) but the plots obtained from measured data. This must be clarified in the text. However, a better option could be to include the data samples in zip files, since this facilitates the reproducibility

Response 4: We will upload statistical data in Appendix C.

Reviewer 2 Report

-          Line 127: Is the OPAL IMU previously validated with a gold standard or other sensors?

-          Line 129: Please define in a precise manner the allocation of the sensors

-          Line 132: Could you add the sensor precision?

-          Line 136: Lenght of walking test and instructions to participants.

-          Line 153: Only one walk test? Or do you measure 2 or 3 times and get the mean value?

-          Figure 3: Whats is “omega”. Figure B is correlated with D in time and space. It´s the same subjects and in the same timestamp?

-          Linea 165: “Clinically, gait….” Spelling error.

-          Asym sp  and JGait and other formula have a reference number, however, I don´t see the reference.

-          Line 197: Please add the software used for Statistical analyses.

-          Table 2: Add BMI and comparison between both groups

-          Table 3; distance of Normal Walk? And for Tandem? Also, you could add the walking speed.

-          Line 368. You have data for analysis of the postural position in dynamic way. You can measure Jwait and Jcehst for walking and analyze the deviations.

Author Response

Point 1:  Is the OPAL IMU previously validated with a gold standard or other sensors?

Response 1: According to our reference 22 (Khobkhun, F.; Hollands, M. A.; Richards, J.; Ajjimaporn, A., Can We Accurately Measure Axial Segment Coordination during Turning Using Inertial Measurement Units (IMUs)? Sensors (Basel) 2020, 20, (9).), this study suggests that the IMU sensors provide a viable alternative to camera-based motion capture that could be used in isolation to gather data from individuals with movement disorders in clinical settings and real-life situations.

Point 2: Line 129: Please define in a precise manner the allocation of the sensors

Response 2: Four IMUs were attached to the subjects: two on the middle of the shanks at midpoint between patellar level and lateral malleolus level, one on the waist at anterior superior iliac spine(ASIS) level, and one on the chest at scapular spine level, as shown in Figure 2. These IMUs recorded the kinematic data of the subjects at a sampling rate of 128 Hz.

Point 3: Line 132: Could you add the sensor precision?

Response 3: We cited this article “Validity and repeatability of inertial measurement units for measuring gait parameters(Washabaugh, et, 2017) by using Lin’s concordance correlation coefficients (LCC), and showed that APDM IMU sensor is accurate and repeatable for measuring spatiotemporal gait parameters in healthy young adults.

Point 4: Line 136: Length of walking test and instructions to participants.

Response 4: 20 meters

Point 5: Line 153: Only one walk test? Or do you measure 2 or 3 times and get the mean value?

Response 5: They were required to perform this walk for 2 times.

Point 6: Figure 3: What is “omega”. Figure B is correlated with D in time and space. It´s the same subjects and in the same timestamp?

Response 6: ωγ represents angular velocity recorded by IMU within a gait cycle

Point 7: Linea 165: “Clinically, gait….” Spelling error.

Response 7: The manuscript text is revised.

Point 8: Asym sp  and JGait and other formula have a reference number, however, I don´t see the reference.

Response 8: The reference will be cited in the revised text.

 Superior gait performance and balance ability in Latin dancers(Liu,et.2022)

Point 9: Line 197: Please add the software used for Statistical analyses.

Response 9: The manuscript text will be revised by adding the following information:

The measured data are presented as mean and standard deviation (mean ± SD). We applied the Shapiro-Wilk test to assess the distribution of data. The overall difference between the groups was tested using Student’s t-test in the case of normal data distribution or nonparametric Mann-Whitney U test when the data were not normally distributed. Software of Microsoft Office Excel 2019 was used for IMU data analysis. Statistical significance was set at < 0.05. A priori sample estimate was calculated using a large effect size (0.82) with the G*Power software (version 3.1; Kiel University, Kiel, Germany), and it was estimated that a minimum of 25 participants per group would be required to detect group differ-ence in gait asymmetry using an alpha of 0.05 and a beta of 0.2.

Point 10: Table 2: Add BMI and comparison between both groups

Response 10: The BMI data is added in the manuscript.

Point 11: Table 3; distance of Normal Walk? And for Tandem? Also, you could add the walking speed.

Response 11: Distance of normal and tandem walk: 20 meters. Walking speed is based on participants’ most comfortable pace.

Point 12: Line 368. You have data for analysis of the postural position in dynamic way. You can measure Jwaist and Jchest for walking and analyze the deviations.

Response 12: Jwaist and Jchest are parameters we use to measure the scale of swaying of our participants during single leg stance to see if participants are able to maintain stability under this task. In other words, the less swaying, we consider as better postural control. However, we are not sure whether this rule can be applied to dynamic condition, based on the fact that our participants is persistently swaying their trunk during ambulation, therefore larger Jwaist and Jchest may not indicate poorer balance control.

Reviewer 3 Report

In this study, the authors investigated the gait symmetry and the single-stance balance in two cohorts of professional yoga instructors and typically developed adults using IMUs. They reported significant differences between the two groups for most of the conditions. Their approach is feasible, and the whole text is well-written in English.

Please find my comments bellow:

#1. The aim and title talk about gait performance which is misleading. Investigating the gait performance, requires comparing various gait parameters (at least the spatiotemporal parameters) which are missing here. Please, either provide gait parameters or rephrase the title and aim.

#2. For better understanding the results, as it is mentioned above, it would helpful to provide spatiotemporal parameters (namely walking speed).

#3. Thanks for providing results for each individual. However, it requires a proper analysis from authors or possible hypothesis for why some participant results is way off comparing to the other (e.g. J Gait   for Y13 and C7 in normal walking Fig.4 a). I believe that would give more value to your work.

#4. Please use conventional gait terms. The terms you use such as ‘gait response’ is quit confusing. 

#5. Introduction L44-69: This paragraph is long and messy. It is hard to read and understand. Half of it just repeated the other half with same citations.

#6. Introduction L78: It is not fully correct. There are lot of global indexes that qualify the gait performances in combination with other factors. With this simple sentence you cannot justified gait performance assessment.

#8.  Introduction L79-82: suddenly authors decided to referencing some EMG and other gait parameters measurement studies. I don’t see the relevant with the whole introduction.

#9. Introduction 103: Please use typically developed instead of normal (in normal control) in the whole text.

#10. Material L107: please use just gait instead of gait behavior.

#11. Material L108: Please refer to the table 2.

#12. Material L136-142: Did authors use a standard protocol for different gait performances and IMU attachment? If so please cite the proper ref and if not please give a short explanation that how you came up with these conditions.

#13.  Material L143-144: “The human gait cycle is symmetrical and periodic; hence, gait performance can be evaluated based on the symmetry of leg movements.” This sentence has been repeated again. Please see my comment #5.

#14.  Material L144-145: a gait cycle typically has 7 events. 1.Heel strike, 2. Loading respond, 3. Mid-stance, 4. Terminal stance. 5. Pre-swing, 6. Mid-swing and 7. Terminal swing.

#15.  Material L146: “MS occurs at the maximum angular velocity of the shank …” the sentence order is not right. “The maximal angular velocity of the shank occurs at MS…”

#16.  Material L155: “The gait responses were further divided into gait cycles according to MS events, as illutrated in Figures 3(d) and 3(e). “

What is gait responses? You mean gait data? You mean ‘walking data was normalized to a gait cycle?’ A gait cycle is defined from one HS to the next ipsilateral HS. What do you mean according to the MS? How many gait cycles per subject have you used?

Figure 3. what is “left gait response”? you mean left knee angular velocity during walking? What about the units? In Fig b and c the units are rad/sec in the d and e are deg/sec (°/sec) which one is right?

#17. Formula 1: What is P?

#18 Material L165-169: these lines belong to the introduction or discussion. By ‘linically’ you mean ‘Clinically’?

#19 Material 183: “…but they were free to stand on both feet if their body lost balance.” Please report how many subjects lost their balance in the results. That would be valuable information.

#20 Results: L218: “The swing time on both sides accounts for approximately 40% of the complete gait” You almost repeated this sentence more than 4 times in the text.

#21 Table 3: What is gait parameter? You Mean J gait? As I understood Jgait according to the formula 1 is dimensionless so what is ‘sec’ mean here?

#22 Figure 4. These are average Asymsp or J gait. By average you mean Average across gait cycles for each subject? Isn’t Jgait supposed to do the same?

#23 Results L259: Please use Pvalue <0.001 instead of 0.00

#24 Discussion: There are several good points discussed in the discussion section. However, concluding several claims from only three IMU outcomes is not enough. The authors need to provide more gait data (kinetics, kinematics and EMG) to support their claims. I would suggest the discussion rephrased more cautiously.

Author Response

Point 1: The aim and title talk about gait performance which is misleading. Investigating the gait performance, requires comparing various gait parameters (at least the spatiotemporal parameters) which are missing here. Please, either provide gait parameters or rephrase the title and aim.

Response 1: The title will be adjusted from gait performance to gait symmetry.

Point 2: For better understanding the results, as it is mentioned above, it would helpful to provide spatiotemporal parameters (namely walking speed).

Response 2: We focused on gait symmetry in this study, thus we consider adjusting the focus and aim.

Point 3: Thanks for providing results for each individual. However, it requires a proper analysis from authors or possible hypothesis for why some participant results is way off comparing to the other (e.g. J Gait   for Y13 and C7 in normal walking Fig.4 a). I believe that would give more value to your work.

Response 3: Thank you for your patience in examining these statistics. Indeed we have noticed that some participants showed inferior performance compared to others, and we have already excluded musculoskeletal disorders or neurological disorders. Those with sedentary lifestyle are also excluded as well. The only possible explanation for poorer performance of the subjects mentioned above may be related to less core muscle training and restrictive exercise.

Point 4: Please use conventional gait terms. The terms you use such as ‘gait response’ is quit confusing. 

Response 4: The text is re-written under your suggestion.

Point 5: Introduction L44-69: This paragraph is long and messy. It is hard to read and understand. Half of it just repeated the other half with same citations.

Response 5: The introduction is largely revised and simplified.

Point 6: Introduction L78: It is not fully correct. There are lot of global indexes that qualify the gait performances in combination with other factors. With this simple sentence you cannot justified gait performance assessment.

Response 6: We have revised gait performance assessment to a parameter of gait stability.

Point 8:  Introduction L79-82: suddenly authors decided to referencing some EMG and other gait parameters measurement studies. I don’t see the relevant with the whole introduction.

Response 8: This reference is removed from the manuscript.

Point 9: Introduction 103: Please use typically developed instead of normal (in normal control) in the whole text.

Response 9: The manuscript is revised as suggested.

Point 10: Material L107: please use just gait instead of gait behavior.

Response 10: The manuscript is revised as suggested.

Point 11: Material L108: Please refer to the table 2.

Response 11: The manuscript is revised as suggested.

Point 12: Material L136-142: Did authors use a standard protocol for different gait performances and IMU attachment? If so please cite the proper ref and if not please give a short explanation that how you came up with these conditions.

Response 12: The citation is added to the manuscript.

Point 13:  Material L143-144: “The human gait cycle is symmetrical and periodic; hence, gait performance can be evaluated based on the symmetry of leg movements.” This sentence has been repeated again. Please see my comment #5.

Response 13: The manuscript is revised as suggested.

Point 14:  Material L144-145: a gait cycle typically has 7 events. 1.Heel strike, 2. Loading respond, 3. Mid-stance, 4. Terminal stance. 5. Pre-swing, 6. Mid-swing and 7. Terminal swing.

Response 14: This is added into our manuscript and we focus on three important gait events, as shown in Figure 3(a): mid-swing (MS), heel strike (HS), and toe-off (TO).

Point 15: Material L146: “MS occurs at the maximum angular velocity of the shank …” the sentence order is not right. “The maximal angular velocity of the shank occurs at MS…”

Response 15: The manuscript is revised as suggested

Point 16:  Material L155: “The gait responses were further divided into gait cycles according to MS events, as illutrated in Figures 3(d) and 3(e). “

What is gait responses? You mean gait data? You mean ‘walking data was normalized to a gait cycle?’ A gait cycle is defined from one HS to the next ipsilateral HS. What do you mean according to the MS? How many gait cycles per subject have you used?

Figure 3. what is “left gait response”? you mean left knee angular velocity during walking? What about the units? In Fig b and c the units are rad/sec in the d and e are deg/sec (°/sec) which one is right?

Response 16: This paragraph is re-written and gait response represents angular velocity.

Point 17: Formula 1: What is P?

Response 17: P is the percentage of swing phases on the right and left feet, respectively, mentioned below the Formula.

Point 18: Material L165-169: these lines belong to the introduction or discussion. By ‘linically’ you mean ‘Clinically’?

Response 18: This error will be revised as suggested, thank you.

Point 19: Material 183: “…but they were free to stand on both feet if their body lost balance.” Please report how many subjects lost their balance in the results. That would be valuable information.

Response 19: The data is provided at Appendix B.

Point 20: Results: L218: “The swing time on both sides accounts for approximately 40% of the complete gait” You almost repeated this sentence more than 4 times in the text.

Response 20: This line will be removed in the revised version.

Point 21 Table 3: What is gait parameter? You Mean J gait? As I understood Jgait according to the formula 1 is dimensionless so what is ‘sec’ mean here?

Response 21 : Gait parameter is J gait in this paragraph. ‘sec’ will be removed in the revised version.

Point 22 Figure 4. These are average Asymsp or J gait. By average you mean Average across gait cycles for each subject? Isn’t Jgait supposed to do the same?

Response 22: J gait represents root mean square average of an individual’s asymmetry of gait. In this figure, it shows the distribution of J gait among our participants. In other words, this is not a figure presenting a single individual’s each gait asymmetry, instead it presents all participants’ J gait value.

Point 23 Results L259: Please use Pvalue <0.001 instead of 0.00

Response 23: This is a mistake and will be revised as suggested.

Point 24 Discussion: There are several good points discussed in the discussion section. However, concluding several claims from only three IMU outcomes is not enough. The authors need to provide more gait data (kinetics, kinematics and EMG) to support their claims. I would suggest the discussion rephrased more cautiously.

Response 24: We have revised gait performance to gait symmetry based on the results of this study.

Round 2

Reviewer 2 Report

Thanks for the modification. Accepted in present form.

Reviewer 3 Report

Thanks to authors for addressing my comments. I have no more comments.